# A Review of Sturge–Weber Syndrome Brain Involvement, Cannabidiol Treatment and Molecular Pathways

**DOI:** 10.3390/molecules29225279

**Published:** 2024-11-08

**Authors:** Katharine Elizabeth Joslyn, Nicholas Flinn Truver, Anne Marie Comi

**Affiliations:** 1Department of Neurology and Developmental Medicine, Hugo Moser Kennedy Krieger Research Institute, Baltimore, MD 21205, USA; joslynk@kennedykrieger.org (K.E.J.); truver@kennedykrieger.org (N.F.T.); 2Department of Neurology, Johns Hopkins School of Medicine, Baltimore, MD 21205, USA; 3Department of Pediatrics, Johns Hopkins School of Medicine, Baltimore, MD 21205, USA

**Keywords:** Sturge–Weber syndrome, cannabidiol, epilepsy, port-wine birthmark, stroke, *GNAQ* mutation

## Abstract

Sturge–Weber syndrome (SWS) is a rare congenital neurocutaneous disorder typically caused by a somatic mosaic mutation in R183Q *GNAQ*. At-risk children present at birth with a capillary malformation port-wine birthmark. The primary diagnostic characteristic of the disorder includes leptomeningeal enhancement of the brain, which demonstrates abnormal blood vessels and results in impaired venous drainage and impaired local cerebral perfusion. Impaired cerebral blood flow is complicated by seizures resulting in strokes, hemiparesis and visual field deficits, hormonal deficiencies, behavioral impairments, and intellectual disability. Therefore, anti-seizure medication in combination with low-dose aspirin is a common therapeutic treatment strategy. Recently published data indicate that the underlying mutation in endothelial cells results in the hyperactivation of downstream pathways and impairment of the blood–brain barrier. Cannabidiol (CBD) has been used to treat medically refractory seizures in SWS due to its anti-seizure, anti-inflammatory, and neuroprotective properties. Pilot research suggests that CBD improves cognitive impairment, emotional regulation, and quality of life in patients with SWS. Recent preclinical studies also suggest overlapping molecular pathways in SWS and in CBD, suggesting that CBD may be uniquely effective for SWS brain involvement. This review aims to summarize early data on CBD’s efficacy for preventing and treating epilepsy and neuro-cognitive impairments in patients with SWS, likely molecular pathways impacted, and provide insights for future translational research to improve clinical treatment for patients with SWS.

## 1. Introduction

Sturge–Weber syndrome (SWS) is a rare, non-inherited neurocutaneous disorder caused by a somatic mosaic mutation in R183Q *GNAQ* [1]. Capillary–venous malformations of the skin, brain, and eyes are key features of SWS, causing many downstream abnormalities: a port wine birthmark of the face, seizures, stroke and brain injury, visual impairment and neglect, glaucoma, hemiparesis, hormonal deficiencies, intellectual disability, and headaches [2]. While the R183Q *GNAQ* mutation occurs early in embryonic development, the neurological presentations usually first manifest in infancy. Seizure activity in SWS tends to have onset during the first two years of life, with stabilization of seizure frequency and duration often occurring around early childhood (~5 years old) [3].

Around 75% of patients with SWS brain involvement have convulsive seizures [4]. Seizure presentation in SWS varies depending on the region and extent of brain involvement, age of seizure onset, and control of seizures and stroke-like episodes achieved. In infants, seizure activity is typically focal with impaired consciousness and includes a motor component, manifesting as twitching of the extremities, eye deviation, and nystagmus, followed by low muscle tone or weakness [2]. If properly addressed, seizure frequency and duration often stabilize by early childhood. For some patients with SWS, the recommended combination of anti-seizure medications and low-dose aspirin does not lead to controlled seizures. Patients may undergo neurosurgical options, including hemispherotomy or resection or disconnection of the cortical region of the brain where the seizure activity is occurring. The efficacy, tolerability, and safety of other forms of anti-seizure treatment, such as the mammalian target of rapamycin (mTOR) inhibitors, like rapamycin (Sirolimus), have been researched in other pediatric seizure disorders, as well as in SWS [5]. Recent pilot studies suggest highly purified cannabidiol (Epidiolex) may be an efficacious method of aiding in medically refractory seizures, as well as in improving cognitive, psychiatric, and neurological outcomes in patients with well-controlled seizures [6,7].

This review aims to address the clinical and molecular features of SWS, as well as the neuroprotective and anti-epileptic properties of cannabidiol (CBD) and molecular effects of CBD that may explain the positive outcomes recently observed in recent pilot trials treating seizures and cognitive impairments in patients with SWS. Future directions regarding research on CBD and SWS are also discussed.

## 2. Clinical Presentation

### 2.1. Skin and Eye Involvement

Due to the somatic mosaic nature of SWS during early fetal development, a facial port-wine birthmark (PWB) is a common feature of the disorder. Early laser surgery during infancy is typically recommended to reduce the vascularity of the birthmark and prevent tissue hypertrophy, blebbing, and bleeding, which frequently happens later in childhood or adulthood in these patients [8]. The PWB mainly presents ipsilateral on the face in relation to the location of brain involvement [2]. Patients with a PWB are susceptible to other dermatological complications, such as excessive nose or gum bleeds. The *GNAQ* mutation is considered a driver mutation, making affected tissues particularly vulnerable to other mutations and vascular proliferation [9]. Presentation of a PWB at birth is one of the early indicators of SWS; however, it is possible for a patient to have brain involvement and not have a PWB. Patients with unilateral PWB presentation have a ~25% risk of increased SWS symptom severity, and those with bilateral presentation have a ~35% risk of symptom severity [10]. Many patients with SWS have a PWB that also covers the upper and lower eyelids, often indicating impaired ocular vasculature and a high risk of glaucoma (Figure 1c) [2].

Congenital glaucoma and open-angle glaucoma are the two most common ocular complications in SWS [11]. While the exact pathophysiology underlying glaucoma manifestation in SWS is still unclear, increased intraocular pressure is most likely due to the *GNAQ* mutation’s impact upon developing vascular cells. A total of 60% of infants with SWS and glaucoma present early, and 40% develop glaucoma later in childhood, teen years, or adulthood [12]. Infants with SWS and glaucoma are at risk for developing buphthalmos, cataracts, iris heterochromia, and vision loss, which can happen at any age with glaucoma. The current first line of treatment for glaucoma in SWS is daily eye drops, which reduce fluid production in the eye or improve outflow. In those with severe glaucoma, valve implant surgery may be performed to aid in decreasing intraocular pressure and increasing aqueous humor outflow [13]. Other surgical procedures, such as goniotomy and trabeculotomy, are also commonly conducted in cases of severe glaucoma [14].

### 2.2. Brain Involvement

SWS brain involvement is seen in Type 1 (brain and skin, with or without eye involvement) and in Type 3 (isolated brain involvement). Since SWS is a spectrum disorder, it is clinically most useful for patients when clinicians refer to the structures involved (or at risk) rather than using the type classification. The same gene mutation has been shown to be active in both Type 1 and Type 3; the difference is likely in the timing of the mutation during fetal development, impacting structures and cell types involved [1]. The predominant diagnostic criterion of SWS brain involvement is visualizing the leptomeningeal angiomatosis. Identified on T1-weighted post-contrast magnetic resonance imaging (MRI), leptomeningeal enhancement indicates capillary–vascular malformations of the leptomeninges and impaired superficial venous drainage pathways in the brain [15]. Other abnormalities that arise over time include cortical and subcortical calcification surrounding blood vessels, as well as the development of enlarged and/or increased deep-draining vessels [16]. Increased prominence of deep-draining vessels acts to compensate for the elevated venous pressure caused by the capillary–venous malformations of the leptomeninges [17]. Further, enlargement of the choroid plexus, commonly observed on T1-weighted post-contrast or susceptibility-weighted MRI, can also develop due to venous hypertension and increased flow in the deep venous system resulting from the leptomeningeal vascular malformation [18]. Observation of peri-vascular brain calcification and atrophy on head computed tomography or susceptibility-weighted imaging on MRI demonstrates the brain injury that develops over time in many patients with SWS (Figure 1a,b) [19]. Bilateral brain involvement is present in around 15% of patients with SWS, and this subset of patients with SWS usually experience more severe seizures and neurocognitive deficits [20].

### 2.3. Seizures and Other Acute Neurologic Crises in SWS

A culmination of neurovascular insults in patients with SWS contributes to focal epilepsy. Seizures are present in 75% of patients with SWS who have unilateral brain involvement and 95% who have bilateral brain involvement [21]. Seizures in infants and young children with SWS are most commonly focal motor with impaired consciousness, and seizure episodes are frequently prolonged, repeated, and severe. Electroencephalography (EEG) is often used to identify seizure activity in patients with SWS. Moreover, quantitative EEG (qEEG) is used to understand and detect abnormalities in the frequency and duration of brain activity in patients with SWS. Asymmetrical electrical brain activity can be seen on qEEG in epileptic patients with SWS, typically appearing as slowed waveforms on the hemisphere with more brain involvement [22]. Seizure activity in SWS contributes to hypoperfusion, and depending on the extent of brain involvement, stroke and stroke-like episodes can arise [23,24].

While stroke-like episodes are a common clinical feature of SWS, their pathophysiology remains poorly understood as they do not present as a typical ischemic infarct on MRI [25]. The most recent study on stroke-like episodes, or transient episodes of hemiparesis, found that most of their cohort (54.5%) had associations with seizures or status epilepticus prior to acquiring permanent or transient hemiparesis; other associations were with head injury, surgery, or unknown. Stroke-like episodes can present similarly as periods of hemiparesis, blurry vision, headaches/migraines, and sometimes impaired speech, usually ranging in time length to fully recover [24]. Results from recent studies on seizures and stroke-like episodes in SWS suggest the importance of delaying or preventing seizure onset and occurrence to prevent injury to the brain, as well as to promote the development of compensatory venous drainage through the deep venous system [26,27]. These patients frequently have prolonged events with a combination of seizure, stroke-like episodes, and migraine-like symptoms, which can result in hospitalization, intensive care unit stays, and intensive therapy needs [28]; these events can be referred to as acute neurologic crises. Acute neurologic crises in SWS are frequently triggered by fever and illness, suggesting that factors such as temperature, inflammation, and dehydration have a role in triggering seizures, stroke-like symptoms, and migrainous symptoms [29].

### 2.4. Cognitive and Psychiatric Outcomes

The cognitive and psychiatric presentations in patients with SWS are highly variable. The extent of brain involvement correlates with the extent of cognitive impairment. Further, the age of seizure onset, the degree of seizure severity, and whether or not seizures are well-controlled from the time of onset are other indicators of altered cognitive development [30]. Common neuropsychiatric diagnoses seen in SWS include Attention Deficit Hyperactivity Disorder (ADHD), anxiety, and Autism Spectrum Disorder (ASD) [31,32,33]. Moreover, motor weakness and dysfunction, as well as visuospatial impairments and learning disabilities or cognitive impairment, frequently arise in patients with SWS [34].

The National Institute of Health’s (NIH) Pediatric Quality of Life in Neurological Disorders (Neuro-QoL) has been used to better understand how SWS symptomatology impacts different aspects of patients’ quality of life. A study including 22 patients with SWS (average age of seizure onset was 2.75 ± 0.99 SEM) collected Neuro-QoL scores and observed a significantly lower cognitive functioning Neuro-QoL subscore, which was associated with the following factors: male gender, extensive SWS brain, skin, and eye involvement, younger age of seizure onset, and antidepressant usage [35]. Routine neurological and neuropsychological evaluation is imperative for tracking the progression of symptom severity and observing areas in patients’ lives that may require more attention, resources, and care. Neurologic outcomes of SWS in youth and adolescents have been a primary focus in the field; however, little is still known regarding the natural history of SWS as patients age. Further research in this area would be beneficial for understanding both the progression of brain injury and seizure severity and how they impact neuropsychological outcomes in adulthood [33].

## 3. Seizure Treatment

### 3.1. Anti-Seizure Medications and Low-Dose ASPIRIN

Anti-seizure medications, most commonly oxcarbazepine and levetiracetam, in combination with low-dose aspirin, are commonly administered for treating seizures in patients with SWS [36]. Other anti-seizure medications used in these patients (carbamazepine, phosphenytoin, topiramate, valproic acid, and lamotrigine) act on a variety of ion channels to downregulate neuronal excitation (Figure 2) [37]. Other drugs (clobazam, phenobarbital, and other benzodiazepines) act to upregulate GABAergic receptors to increase inhibitory signaling and control the over-excitation that leads to seizures in the epileptic brain [38]. These medications can have diffuse side effects ranging from movement disorders, dermatological reactions, visual side effects, weight and metabolic changes, and other complications [39]. Altered cognitive functioning is another common side effect of anti-seizure medications. Studies have found associations between topiramate and impairments in concentration and cognitive performance. Other studies have found side effects of drowsiness in oxcarbazepine, as well as emotional regulation issues in levetiracetam [40]. Cannabidiol/Epidiolex is a newer option, which has been helpful for these patients and modulates brain activity through a variety of mechanisms (Figure 2). The ketogenic or Atkins diets can improve seizure control but are difficult to maintain in the long term. For older children and adults who have failed medical management, a vagus nerve stimulator (VNS) can be helpful. Surgical resection or disconnection to the region where seizure activity is occurring is the best method of addressing medically refractory epilepsy in patients who are deemed to be good candidates [2].

Low-dose aspirin therapy for SWS aims to increase brain blood flow and mitigate stroke and stroke-like episodes that are commonly present in SWS [41,42]. A study looking at the clinical outcomes of aspirin use in patients with SWS found that 84% of their study population had no reported side effects while on low-dose aspirin. Minor complications, like increased bruising or nosebleeds, were the primary reported side effects. Improvements in academics and motor abilities were also found in patients; however, all the patients with bilateral brain involvement had consistently low cognitive functioning neurologic scores, even while on low-dose aspirin. Overall, most subjects had “good” cognitive functioning (80%) and final seizure (91%) scores [42]. Anti-seizure medications in combination with low-dose aspirin can be successful in reducing both stroke-like episodes and seizure frequency and duration; however, a significant subset of patients with SWS experience medical refractory seizures in which their seizures do not subside after trying multiple first-line anti-epileptic medications.

**Figure 2 molecules-29-05279-f002:**
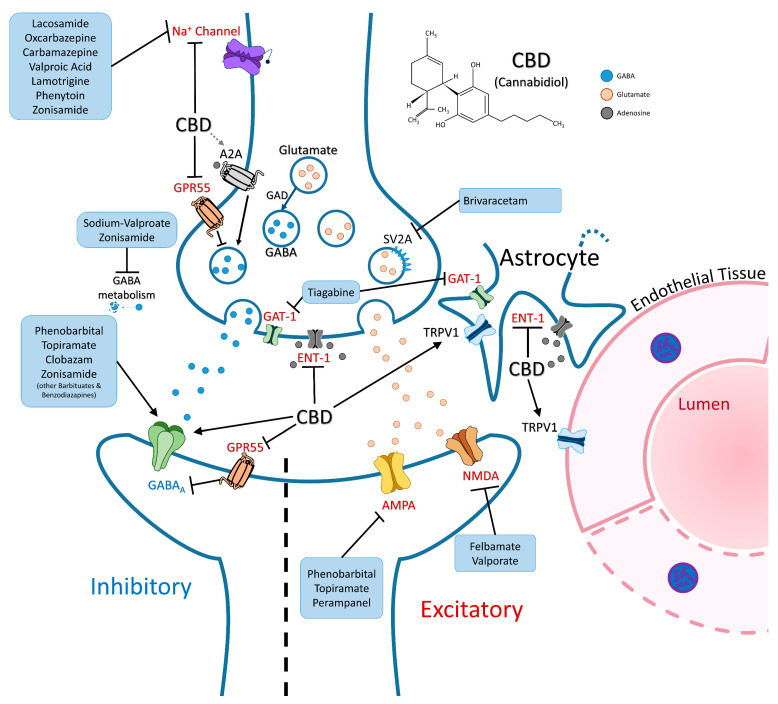
Diagram of cannabidiol in the context of common anti-seizure medications prescribed in SWS (not to scale). Overall, CBD has a pro-inhibitory effect via various receptor pathways. As proposed by Rosenberg et al. [43], GPR55 promotes the downregulation of GABA_A_ receptor densities via the breakdown of gephyrin scaffolding; however, this effect is blocked by CBD. Activation of TRPV1 by CBD may regulate vasodilation and astrocytic movement post-seizure. Additionally, the regulation of the adenosine transporter (ENT-1) may play a bigger role in SWS by promoting both GABA release via A2A and AMPA endocytosis via A1. Adapted from Löscher and Klein 2021 [44]. Abbreviations: CBD (Cannabidiol); GABA (gamma-aminobutyric acid); GABA_A_ (gamma-aminobutyric acid receptor A); AMPA (α-amino-3-hydroxy-5-methyl-4-isoxazolepropionic acid receptor); NMDA (N-methyl-D-aspartate receptor); TRPV1 (transient receptor potential vanilloid 1); GAD (glutamic acid decarboxylase); GAT-1 (gamma-aminobutyric acid transporter 1); A1 (adenosine A1 receptor); A2A (adenosine receptor subtype 2A); ENT-1 (equilibrative nucleoside transporter 1); GPR55 (G-protein coupled receptor 55); SV2A (Synaptic vesicle glycoprotein 2A).

### 3.2. CBD and Pilot Studies for SWS

Highly purified CBD (Epidiolex) as an adjunctive medicinal treatment has been studied in medically refractory pediatric epilepsy syndromes and is FDA-approved for use in Dravet syndrome and Lennox–Gastaut syndrome [45]. A recent pilot study found therapeutic benefits of CBD (5–25 mg/kg/day) in patients with SWS who had medically refractory seizures. Seizure frequency significantly decreased in four out of five subjects, and most participants reported subjective improvements in cognitive, behavioral, and motor functioning, as well as quality of life. Three of the participants in this study also had bilateral brain involvement and failed 2–7 previous anti-epileptic medications; however, after completion of the six-month study, each had over a 50% decrease in seizure frequency at their most recent visit while taking CBD, compared to baseline. In these three participants, no significant differences in rescue medication use were noted between baseline and time on CBD. Additionally, these participants remained on CBD for at least 63–80 weeks in the extension phase [7].

The impact of CBD on patient cognition, mood, and behavior has also been researched in patients with SWS. A follow-up pilot study was recently conducted looking at the neurological, cognitive, and psychiatric outcomes in ten subjects with SWS and well-controlled seizures who had been administered CBD. As opposed to the previous study, they utilized neuropsychological, neuropsychiatric, and developmental examinations to objectively measure improvements. During the 6-month study, there were no reported seizures. Based on scores using tests like the Behavioral Assessment for Children 3rd Edition (BASC)-3, Screen for Child Anxiety Related Disorders (SCARED), and Behavior Rating Inventory of Executive Functioning (BRIEF)-2 scale, Smegal et al. observed improvements in emotion regulation and anxiety. They further found significant improvements in motor functioning using scores on the ABILIHAND-Kids, as well as overall neurological improvements on the SWS neuroscore [6]. These pilot studies suggest that in patients with SWS, with either medically refractory seizures or with cognitive impairments, adjunctive medicinal treatment with CBD/Epidiolex may be beneficial [6,7]. Further studies are needed to assess the long-term safety, tolerability, and efficacy of CBD/Epidiolex in patients with SWS.

## 4. *GNAQ* Mutation and Molecular Pathways Implicated

### 4.1. Somatic Mutation

SWS, as well as non-syndromic facial PWBs, are both typically caused by a somatic mosaic mutation in R183Q in the *GNAQ* gene on chromosome 9q21 [1]. Because the mutation likely happens in a subset of angioblasts during fetal development, the mutation is localized to vasculature and tissues that surround the affected angioblasts of the developing eye, brain, and skin tissues. It is theorized that the extent of the tissues affected is dependent on when, during development, the missense mutation happens in a progenitor cell [1]. If an early progenitor cell is affected, the mutation could involve the adjacent nascent brain, eye, and skin tissues; if the mutation occurs later, when those tissues are more developed and more spatially separated, then only one tissue is affected (i.e., skin or brain tissue) [42]. It is important to note that mutation of the *GNAQ* gene happens after the zygote is formed, does not affect germline cells, and is, therefore, not hereditary. The hallmark study that identified the R183Q *GNAQ* mutation studied brain and skin tissue from patients with SWS, patients with non-syndromic PWB, and patients without PWB or SWS. Among these samples, they found that 100% of patients with SWS had the c.548G>A p. Arg183Gln mutation in port-wine stained skin tissue, and 83% had this same mutation in brain tissue [1]. Other groups subsequently confirmed these results and determined that the R183Q *GNAQ* mutation was enriched in endothelial cells [46]. It is of interest to note that a subset of uveal melanoma, a rare form of eye cancer, occurs when the R183Q *GNAQ* mutation occurs in melanocytes during adulthood rather than in vascular cells during fetal development, as in SWS and PWB [47].

### 4.2. Normal Functioning Canonical GNAQ/Gαq Pathway

The *GNAQ* gene is responsible for encoding the alpha subunit of the q class G protein (G_αq_) of metabotropic G-protein coupled receptors (GPCRs) (Figure 3). G_q_ GPCRs make up a plethora of receptors in both neuronal and endothelial tissues, such as the α1 adrenergic receptor (A1), angiotensin 2 receptor type 1 (AT1), or the endothelin 1 receptor. The c.548G>A p. Arg183Gln mutation happens in the Switch I region of the G_αq_ protein, which contains the binding region for guanosine triphosphate or guanosine diphosphate (GTP/GDP) as well as the Ras-like catalytic region that is involved in GTP hydrolysis [48]. On its own, G_αq_ is a weak hydrolase of GTP (GTPase), so binding and activating phospholipase C beta (PLCβ) is needed in order to facilitate GTP hydrolysis [49].

Normally, membrane-bound heterotrimeric G_q_ protein GPCR activation by its ligand causes the GDP-bound α subunit to separate from the gamma/beta subunits. The α subunit replaces the GDP for GTP, which causes a conformational change in the switch I and switch II regions of the subunit, effectively ‘activating’ the subunit. The G_αq_ protein then binds to and is able to activate PLCβ, which in turn cleaves phosphatidylinositol 4,5-bisphosphate (PIP2) into diacylglycerol (DAG) and inositol trisphosphate (IP3) [50]. PLCβ then facilitates the GTPase activity of the G_α_ protein, which returns the α subunit back to the ‘inactive’ GDP-bound state, allowing it to reform with the heterotrimeric subunit [48]. IP3 will pass through the cytosol to activate ionotropic IP3 receptors on the surface of the endoplasmic reticulum that release calcium intracellularly [51]. DAG will activate protein kinase C (PKC), which activates the mitogen-activated protein kinase (MAPK) pathway among other targets [46]. Increased expression of phosphorylated extracellular protein kinase (p-ERK) can increase the mammalian target of rapamycin (mTOR) activity (Figure 3). Thus, G_q_ proteins directly interact with the PLC/IP3/DAG pathway but can also influence the MAPK and mTOR pathways [2,48]. 

### 4.3. Mechanisms of GNAQ and Other Related Mutations

Once the active G_αq_ unbinds from PLCβ, it can then consecutively activate PLCβ and cause over-activation of further downstream effects. More recent research has revealed that other, rarer gene mutations have been associated with SWS, including mutations in the *GNB2* (c.232A>G, p.Lys78Glu) and *GNA11* (c.547C>T, p.Arg183Cys) genes [52,53]. These somatic mutations appear phenotypically similar to the *GNAQ* mutation and have similar molecular etiology. The *GNA11* gene codes for the production of the alpha subunit of the G_11_ heterotrimeric protein, which is a homolog of the G_q_ protein. The *GNB2* gene codes for the beta subunit of the G protein. The *GNA11* mutation, like the *GNAQ* mutation, is also implicated in uveal melanoma in adults [54]. Studies of these mutations suggest the involvement of abnormal Hippo signaling pathways in SWS. Implicated in cell proliferation, organ growth and development, and tissue growth restriction, the hippo pathway is a kinase cascade involving actin and other scaffolding proteins. Briefly, GPCRs influence actin dynamics in a Rho-dependent manner, which indirectly inhibits the phosphorylation of yes-associated protein (YAP); YAP will then translocate through the nuclear membrane and promote gene transcription factors [55,56]. In comparisons of tissue samples from the *GNB2* mutation and *GNAQ* mutation, an upregulation of the YAP pathway was seen in both forms of SWS; however, upregulation of mTOR was only found in the *GNAQ* mutation [52]. In studies of uveal melanoma, YAP has been seen to be activated in a manner independent from hippo in the *GNA11* mutation variation of the disease, which suggests a more direct path from the mutated G protein signaling.

## 5. Cannabidiol

### 5.1. CBD Mechanisms of Action and Epilepsy

CBD is one of the two main exogenous phytocannabinoids in the *Cannabis sativa* plant, along with ∆9-tetrahydrocannabinol (THC). Unlike THC, CBD does not have psychoactive properties. Instead of directly activating cannabinoid type-1 receptors (CB1R), CBD has a low binding affinity for CB1R and interacts with a wide range of other receptors that exhibit anti-inflammatory, neuroprotective, and anti-epileptic effects. The two main endogenous ligands of the cannabinoid receptors are 2-arachidonoylglyercol (2-AG) and N-arachidonoylethanolamine (AEA). These endocannabinoids are important in the endocannabinoid system (ECS) and are involved in a variety of regulatory processes involving sleep, immunity, learning, memory, temperature, pain, and emotions [57].

Famously, a strain of CBD-rich cannabis colloquially known as ‘Charlotte’s Web’ was discovered to attenuate seizures in a case study of Dravet Syndrome in 2014, which led to further clinical trials with Epidiolex, as previously discussed above [58]. The pathophysiology of how CBD attenuated seizures in that case is unknown. CBD acts on many receptors traditionally associated with seizures, such as the gamma-aminobutyric acid (GABA) A receptor [59] and sodium channels [60]; both of these targets are clinically relevant in the management of SWS and seizure mitigation [44,61]. Other probably important pathways include TRPV1, GPR55, and the adenosine receptor-stimulated pathways, which are discussed in more detail below and in Figure 2.

### 5.2. Novel Receptors Modulated by CBD

#### 5.2.1. TRPV1

The transient receptor potential channel subfamily vanilloid (V) member 1 (TRPV1) has been extensively researched in the pathways involving noxious stimuli, such as pain and temperature, and pathways of homeostatic regulation. Recently, the TRPV1 receptor has been shown to be potentially involved in seizure activity given that it is a calcium-permeable ion channel, and the receptor has been seen to be located in hippocampal regions [62]. CBD is known to act as an agonist for the TRPV1 receptor and is able to rapidly desensitize the receptor in vitro [63]. In the normal inactive state, PIP2 is bound to the C-terminus-intracellular domain of the TRPV1 receptor [64]. Subsequent cleavage of PIP2 by PLCβ into IP3 and DAG effectively decreases inhibition of the TRPV1 receptor. This disinhibition could potentially be exacerbated in SWS due to the pathology that leads to the over-activation of PLCβ. Alternatively, activation of the TRPV1 receptor can result from other triggers, including endocannabinoids, heat, inflammatory proteins, low pH, and activation by either PKA or PKC, so it is possible that the ‘unleashing’ of TRPV1 activity in areas that co-localized with mutated G_q_ GPCRs in SWS leads to some of the clinical outcomes [65].

Despite many studies showing that activation of TRPV1 causes neuronal hyperexcitability and seizure-like activity, CBD shows an anticonvulsant effect while still being an agonist for the TRPV1 receptor [66]. Both in vivo and in vitro models show that antagonism of the TRPV1 receptor facilitated attenuation of seizure activity or raised seizure threshold in pentylenetetrazole-induced seizures [67,68]. However, when using a knockout mouse model of TPRV1, the anti-seizure effect of CBD was lessened significantly [69]. This leaves a conundrum about how CBD attenuates hyperexcitability by activating a receptor that promotes hyperexcitability. It may be that CBD acts in a tight dose-dependent manner, as seen in an in vitro study by Anand et al. CBD was found to inhibit TPRV1 at lower doses [70]. Another in vitro study showed that TRPV1 receptors became desensitized after activation by CBD and showed less excitability [71]. It is worth mentioning that a mouse model of Dravet syndrome concluded that TRPV1 was not a viable target for the moderation of seizure activity due to the various age-dependent effects and assumed low TRPV1 expression in younger mice; however, other seizure mouse models have found higher TRPV1 expression in younger mice [72,73].

Another aspect of SWS is the capillary malformations. The mechanism by which this happens is largely unknown; however, it has been theorized to be partially due to a decreased functioning of the inward-rectifying potassium channel 2.1 (Kir2.1). These channels allow small blood vessels to detect neuronal activity, leading to vasodilation [74]. G_αq_-stimulated drops in PIP2 can ‘turn off’ Kir2.1 channels in endothelial cells, possibly causing the malformations [75]. With the normal vasodilation mechanism turned off, TRPV1 activation in endothelial and glial cells could be a path toward rectifying this malformation with CBD. TRPV1 activation has been seen to have a vasodilator effect in brain capillaries (Figure 2) [76].

#### 5.2.2. GRP55

A relatively newly discovered receptor, GPR55 is an endocannabinoid Class A family GPCR. Class A GPCRs are the “Rhodopsin (rho) like” receptors that directly activate the rho-ROCK pathway in the presence of the guanine nucleoside exchange factors (GEF) that facilitate GTPases. Importantly, Rho-like GPCRs are downregulated by GTPase activating proteins (GAPs) and guanine-nucleoside dissociation inhibitors (GDIs). In the case of GPR55, the downstream targets of GTP-bound activated rho can include rho-associated protein kinase (ROCK) and p38-mitogen-activated protein kinase (p38-MAPK) [77]. Endogenously, GPR55 is activated by Lysophosphatidylinositol (LPI), which has been shown in HEK293 cells that express GPR55 [78]. GPR55 is normally found in many brain areas involved in normal seizure pathology, including the hippocampus and the hypothalamus, and it is found upregulated in mouse seizure models [79]. The GPR55 receptor is also found on GABAergic interneurons, which are known to be heavily involved in seizure activity by regulation of the excitatory–inhibitory (E:I) ratio [80].

CBD is an antagonist of the GPR55 receptor, which shows involvement in seizure pathology by effectively increasing GABA transmission. While it has been known that in mouse models, CBD can attenuate seizures in a GPR55-dependent manner, the exact mechanism is still unknown [81]. Rosenberg et al. propose a few mechanisms by which GPR55 effectively inhibits the GABA_A_ receptors on the post-synapse, one of which is via downregulation of the gephyrin scaffolding. This study describes a molecular pathway driven by the rho-ROCK pathway to activate calcineurin, independent of PLC activation, and eventual breakdown of the gephyrin scaffolding [43]. By blocking this breakdown of GABA_A_ receptor clusters, mechanistically, CBD would have an overall effect of upregulating the inhibitory pathway (Figure 2).

Additionally, GPR55 activation in vitro has been seen to activate intracellular calcium influx via G_q_, G_11,_ and rho-dependent mechanisms [82]. These GPR55 receptors have been seen on the presynaptic membrane and are able to activate the release of excitatory transmission [83]. Blockade of this pathway would thus regulate the E:I ratio back to normal levels by CBD.

#### 5.2.3. Adenosine Receptors/Transporter

Adenosine has long been known to modulate overall excitability in the brain, as demonstrated by the popularity of drinking caffeine, which blocks adenosine receptors. Of the adenosine receptors, CBD has been found to modulate the adenosine receptor subtype 2A (A2A) and the adenosine A1 receptor (A1) via the equilibrative nucleoside transporter 1 (ENT-1) [84,85]. A2A and A1 are both GPCRs; however, they activate different G proteins that affect the cyclic adenosine monophosphate (cAMP) pathway. A1 receptors activate G_i/o_ (inhibitory) signaling pathways, while A2A activates G_s_ (stimulating) signaling. Expression of A1 is seen throughout the brain and glial cells, while A2A is seen more in the basal ganglia [86,87,88].

Carrier et al. found that CBD lowered the amount of pro-inflammatory markers by binding and inhibiting ENT-1, which was not the case for CBD-treated A2A knockout mice [84]. Castillo et al. found that A2A meditated this effect in a neonatal mouse cell culture model of hypoxic ischemia [89]. This effect of anti-inflammation by CBD via ENT-1 and A2A has also been shown in the retina in vivo, which is potentially important to SWS in the case of increased intraocular pressure [85]. Implicated in neuronal excitability, the A1 receptor has shown a more inhibitory role and has been found to form heteromers with A2A on glutamatergic presynaptic terminals in the striatum (Figure 2) [90]. Activation of these striatal receptors has been shown to increase GABA signaling; however, the exact pathway is unclear. Activation of A1 has also been seen to facilitate AMPA receptor endocytosis in models of hypoxia [91,92]. As previously mentioned, patients with SWS experience ischemic-like episodes due to venous hypertension and impaired cerebral blood flow exacerbated by seizures. Therefore, CBD may block abnormal inflammatory responses seen in SWS and decrease hyperexcitability by decreasing AMPA receptor activity and increasing GABA signaling [93].

## 6. Conclusions

SWS is a complex disorder that affects the brain, skin, eyes, cognitive functioning, and quality of life in patients diagnosed. Further preclinical and clinical research is needed to better understand how to treat its various clinical implications. Current models of the molecular mechanisms of CBD do demonstrate numerous targets that are key to seizure pathology and control. More preclinical models are needed to elucidate the molecular mechanisms and receptor dynamics behind the effects of CBD-mediated seizure attenuation and neurocognitive improvements in SWS. Understanding CBD’s polymodal actions in SWS will further understand other syndromes associated with these pathways and with seizure pathology. The overlap in impacted molecular pathways and the data from small pilot trials support the continued study of this treatment approach in larger, prospective clinical trials for SWS.

## Figures and Tables

**Figure 1 molecules-29-05279-f001:**
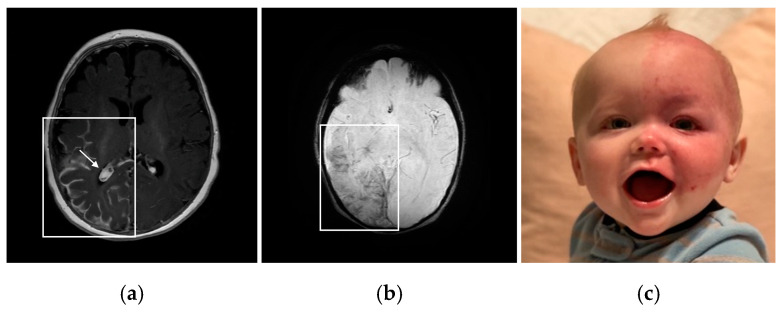
(**a**) Patient 1: T1-weighted post-contrast axial MRI showing unilateral right leptomeningeal enhancement (box) and enlarged choroid plexus (arrow). (**b**) Patient 1: Susceptibility-weighted axial MRI showing right occipital and temporal lobe cortical calcification and dilated deep draining vessels (box). (**c**) Patient 2: Infant with SWS presenting a unilateral left-sided facial PWB and glaucoma.

**Figure 3 molecules-29-05279-f003:**
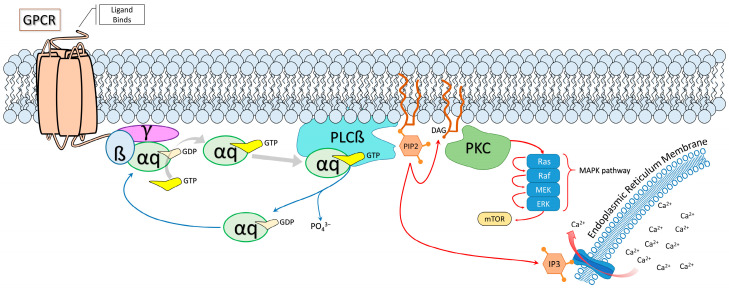
The canonical Gαq pathway is dysregulated by the R183Q somatic mutation that likely causes capillary malformations in the brain, skin, and eye and dysregulation of cellular mechanisms, thereby resulting in symptoms of SWS. Red arrows indicate an upregulated path in the R183Q GNAQ mutation; blue arrows indicate a blocked or downregulated path in the mutation. Since the αq remains in the active form, PLCβ excessively cleaves PIP2 and recursively activates downstream targets. Unidentified are the compensatory changes that happen naturally that mitigate this process, given that SWS is not cancerous. Abbreviations: GPCR (G-protein coupled receptor); GDP (Guanosine diphosphate); GTP (Guanosine triphosphate); PLCß (Phospholipase C beta); PIP2 (Phosphatidylinositol 4,5-bisphosphate); DAG (Diacylglycerol); PKC (Protein kinase C); MEK (Mitogen-activated protein kinase kinase); ERK (Extracellular signal-regulated kinase); MAPK (Mitogen-activated protein kinase); mTOR (mammalian target of rapamycin); IP3 (Inositol trisphosphate).

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
