# Peer review of "A Review of Sturge–Weber Syndrome Brain Involvement, Cannabidiol Treatment and Molecular Pathways"

_molecules, 2024, doi:10.3390/molecules29225279_

Round 1
Reviewer 1 Report
Comments and Suggestions for Authors
The review suggests in good detail how the use of Cannabidiol could help in preventing and treating epilepsyand neuro-cognitive impairments in patients with SWS. Basically, they explain the molecular mechanisms that may be involved.
The investigation into molecular mechanisms gives basics for further laboratory and clinical research .
Therefore, it looks promising but clinical use of CBD still needs further clinical investigation.
Reviewer 2 Report
Comments and Suggestions for Authors
Dear authors,
This paper reviewed the progress of Sturge-Weber syndrome (SWS), cannabidiol (CBD) treatment, and molecular pathways. The review aims to address the clinical and molecular features of SWS, as well as the neuroprotective and anti-epileptic properties of CBD and molecular effects of CBD that may explain the positive outcomes recently observed in recent pilot trials treating seizures and cognitive impairments in patients with SWS. Future directions regarding research on CBD and SWS are also discussed. The review was well written. Some comments and suggestions are listed below.
1. p. 8, line 309: The Latin name “Cannabis sativa” should be in italics.
2. p.10, Figure 3: If the authors refer to other references, please cite them.
3. p.11, section Conclusions: the application of CBD to treat SWS should be prospected.
Author Response
- p. 8, line 309: The Latin name “Cannabis sativa” should be in italics.
Response: Thank you for drawing our attention to this. We have changed the text to italicized to reflect the latin genus and species name for the plant.
- p.10, Figure 3: If the authors refer to other references, please cite them.
Response: We created this figure inspired in part by (but not copying) a Figure in an article from Löscher and Klein 2021. Their citation is now added.
- p.11, section Conclusions: the application of CBD to treat SWS should be prospected.
Response: We have changed the last sentence of the discussion to end with: “…..continued study of this treatment approach in larger, prospective clinical trials for SWS."
Reviewer 3 Report
Comments and Suggestions for Authors
1. Brain involvement is clinically categorized only for Type-3 SWS, do authors make this distinction or the review depicts a general overview of SWS. I think a distinction should be made when it comes to molecular mechanisms; isn’t it?
2. Fig.1. Do authors have the patient’s consent to publish the picture; usually while publishing a patient’s picture a specific format is followed to hide the identity, isn’t it?
3. Section 2.3; what is the order (magnitude) of Seizures brain activity in SWS patients is it distinct or like other forms of seizure activity. Please specify and include to your discussion in section 2.3.
4. In Neuro-QoL study, what came as an average age of seizure onset, if reported please include to the review.
5. Section 3. was the 60% decrease in seizure frequency sustained and is mentioned for which duration in this manuscript, please clarify?
6. Also, during the 60% decrease in seizure frequency; was the amplitude of seizure episodes changed?
7. Section 4.2; and Fig.2. the depiction of pathways; is it in context to seizures or also in context to other SWS-symptoms?
8. Most of the information provided in this review is more relevant for Type-3 SWS, therefore authors may consider a more specific ‘title’ for their manuscript.
9. Please try improving the picture quality.
Comments on the Quality of English Language
10. Please look through the manuscript for typos.
Author Response
1. Brain involvement is clinically categorized only for Type-3 SWS, do authors make this distinction or the review depicts a general overview of SWS. I think a distinction should be made when it comes to molecular mechanisms; isn’t it?
Response: SWS brain involvement is seen in Type 1 (brain and skin, with or without eye involvement) and in Type 3 (isolated brain involvement); SWS is a spectrum disorder and it is clinically most useful for patients when clinicians refer to the structures involved (or at risk), rather than using the type classification. The same gene mutation has been shown to be active in both Type 1 and Type 3; the difference likely is in the timing of the mutation during fetal development impacting structures and cell types involved. This clarification is provided in Section 2.2.
2. Do authors have the patient’s consent to publish the picture; usually while publishing a patient’s picture a specific format is followed to hide the identity, isn’t it?
Response: We have received consent from the patient’s mother, and they have signed the MDPI-specific consent document. The portwine birthmark over the eyelid is a common clinical feature of SWS and is often an indicator of glaucoma, so having the patient’s eyes visible is important to demonstrate this.
3. Section 2.3; what is the order (magnitude) of Seizures brain activity in SWS patients is it distinct or like other forms of seizure activity. Please specify and include to your discussion in section 2.3.
Response: The following statement has been added to this section (second sentence): “Seizures in infants and young children with SWS are most commonly focal motor with impaired consciousness, and seizure episodes are frequently prolonged, repeated, and severe.”
4. In Neuro-QoL study, what came as an average age of seizure onset, if reported please include to the review.
The average age of seizure onset in this group was 2.75 +/- 0.99. This information has been included in the Neuro-QoL section of the manuscript.
5. Section 3. was the 60% decrease in seizure frequency sustained and is mentioned for which duration in this manuscript, please clarify?
Response: The study duration was for six months on cannabidiol, and all three subjects with a >50% reduction in seizure frequency completed the six months study on drug. These three patients remained on cannabidiol for at least 63-80 weeks in the extension phase. This information was added to Section 3.2.
6. Also, during the 60% decrease in seizure frequency; was the amplitude of seizure episodes changed?
Response: This was a small study and the only seizure variable which was quantified and analyzed was seizure frequency. No significant differences in rescue medication use were noted between baseline and time on CBD; this additional piece of information has been added to Section 3.2.
7. Section 4.2; and Fig.2. the depiction of pathways; is it in context to seizures or also in context to other SWS-symptoms?
Response: The pathway is in context of the somatic gene mutation within cells of the capillary malformations in Sturge-Weber syndrome, but may be indirectly related to the seizure activity (resulting from impaired vascular development and function and impaired blood-brain-barrier) that is seen in SWS. We updated the figure 2 description to address this point.
8. Most of the information provided in this review is more relevant for Type-3 SWS, therefore authors may consider a more specific ‘title’ for their manuscript.
Response: We have updated the title to be: A Review of Sturge-Weber syndrome Brain Involvement, Cannadidiol Treatment, and Molecular Pathways to address the primary focus of the review.
9. Please try improving the picture quality.
Response: Thank you. We have gone back and re-saved the images at a higher quality. Figures 2 and 3 should now be sharper at 600 and 417 DPI respectively.